# Impacts of Drought Stress and Mycorrhizal Inoculation on the Performance of Two Spring Wheat Cultivars

**DOI:** 10.3390/plants11172187

**Published:** 2022-08-24

**Authors:** Caroline Pons, Caroline Müller

**Affiliations:** Department of Chemical Ecology, Bielefeld University, 33615 Bielefeld, Germany

**Keywords:** applied water use efficiency, arbuscular mycorrhiza, climate change, drought, harvest index, wheat

## Abstract

Cereal production is becoming challenging, given ongoing climate change. Arbuscular mycorrhizal fungi (AMF) are discussed to mitigate effects of drought for plants and enhance nutrient uptake. Thus, we investigated the impacts of drought and mycorrhiza on the growth and allocation patterns of two cultivars of spring wheat (*Triticum aestivum*). Plants were grown under three irrigation regimes (well-watered, continuous or pulsed drought) and in three substrates (absence or presence of one or three AMF species). Applied water use efficiency (WUE_applied_), harvest index (HI) and contents of carbon (C), nitrogen (N) and phosphorous (P) were determined when grains were watery ripe. When grains were hard, again, WUE_applied_, HI and the thousand-kernel weight were measured. The WUE_applied_ and HI were lowest in plants under pulsed drought stress at the second harvest, while the thousand-kernel weight was lower in mycorrhized compared to non-mycorrhized plants. The C/N ratio dropped with increasing drought stress but was enhanced by mycorrhiza, while the P content was surprisingly unaffected by mycorrhiza. The total root length colonization was higher in substrates with the AMF mix, but overall, fungal presence could not alleviate the effects of drought. Our results highlight the complexity of responses to challenging environments in this highly domesticated species.

## 1. Introduction

Globally, half of all habitable land is already used for agriculture [1] and food demand is still growing [2]. Meanwhile, crop yields have stagnated in several parts of the world [3] and plant growth conditions are becoming challenging due to the impacts of the globally changing climate [4]. Consequences of these changes present themselves as agricultural obstacles, such as soil erosion and depletion, as well as water shortages [5]. The latter manifests in droughts, which are becoming an increasing problem for crop cultivation, impacting plant growth and harvests [6]. Recently, Europe has also been experiencing increasing temperatures and periods of drought during the vegetation period [7,8], resulting partly in remarkable yield losses [9]. Investigating the responses of indispensable crop plants, such as wheat (*Triticum aestivum* L., Poaceae), in terms of growth and yield to different drought conditions is therefore essential.

In general, water shortage limits plant biomass production [10], as has been shown in wheat [11,12,13]. Water-deprived plants close their stomata to avoid further transpiration [14], often enhancing water use efficiency (WUE) [10]. Droughts can occur in different forms, such as continuous droughts, with a constant shortage of water or pulsed droughts, in which longer drought periods are intermitted by heavy rain events. Scenarios in which the stress phases are intermitted may result in an even more severe stress response because the plants have to re-adapt to changing conditions, which can be costly in terms of resource allocation [15]. Under re-watering conditions, drought-stressed plants may invest in new organs, such as leaves or tillers, which cannot be maintained once water is scarce again, thus, organs may be lost in the end [16]. Lower irrigation frequencies can cause a lowered WUE_applied_, altered metabolic responses and a lower harvest index than when plants are kept at higher frequencies of irrigation and a constantly lower soil moisture (i.e., continuous drought) [17,18,19].

To enhance soil properties and nutrient supplies for crop plants, particularly under drought, the maintenance or introduction of arbuscular mycorrhiza (AM), which are produced in a symbiotic relationship between AM fungi (AMF) and plant roots, is a possible approach [20,21,22,23]. If the AMF can be successfully established, the fungal mutualist will provide nutrients such as phosphorous and nitrogenic compounds, as well as water, to the plant. In return, the plant will deliver photo-assimilates to the fungus [22]. Both the belowground fungal C-sink and the improved nutrient supply of the host likely influence C allocation patterns at the whole-plant level. Thus, AM are considered an integral part of the plant’s physiology [24]. Depending on the environmental circumstances, mycorrhized plants can show an enhanced biomass [25,26] and stress resilience compared to non-mycorrhized plants [27,28,29]. AMF are ubiquitous and associate with most vascular plant species [30]. However, in contrast to many forbs, Poaceae species are not obligate mycotrophic [31], and miscellaneous effects of AM on crop species, such as wheat, have been found. Inoculation with different AMFs had no beneficial effect on the biomass of most modern wheat cultivars [32,33] and even had a negative impact on the biomass production of spring wheat [33]. In contrast, some studies revealed positive effects of AM on the biomass production or harvest index of wheat plants [19,34,35]. Higher yields in mycorrhized compared to non-mycorrhized plants were found in two winter wheat cultivars, which were independent of the soil moisture conditions [36]. However, in a field trial with spring wheat (*T. aestivum*), mycorrhizal treatment had an especially positive effect on aboveground biomass and harvest index when plants were exposed to water shortage [37]. Whether AM indeed enhances drought resilience in wheat is still unclear.

In many studies investigating the effects of AMF on plants, often inocula consisting of a single AMF species were used [33,34,38]. However, a natural soil community consists of various AMF and other organisms [39,40], which may have different effects on plant growth and physiology. For example, depending on the genus and species of the AMF, root colonization success and nutrient supply to the plant can differ [41,42]. Thus, the responses of plants to AM establishment may be cultivar-specific, may depend on the AMF species (composition), and are shaped by environmental conditions.

Wheat is a highly domesticated crop that has been selected for homogenously growing plants with high yield [43]. Concurrently, plants of the same genus but different cultivars can show different responses in yield [44] and resistance [45], and can differ in AMF responsiveness [46,47] due to even small genetic variation. In the present study, we investigated the effects of different substrates containing either one (AM1, *Rhizoglomus irregulare*) or a mixture of three AMF species (AM3, *R. irregulare, Funneliformis mosseae, F. caledonium*) versus a non-mycorrhized control (NM) on growth and resource allocation traits of two spring wheat cultivars [Mistral (MIS) and Chamsin (CHA)] exposed to three different irrigation regimes [well-watered control (CTR) versus continuous drought (CD) versus pulsed drought (PD)], using a full-factorial design. Sets of plants were harvested at two time points [T1: 67 days post sowing (dps), grains watery ripe, and T2: 113 dps, grains hard], and hydration status, total root length colonization, WUE_applied_, harvest index and thousand-kernel weight were measured. Moreover, we analyzed the content of carbon (C), nitrogen (N) and elemental phosphorous (P) in the vegetative and reproductive aboveground plant tissues at the first harvest. We expected wheat plants to benefit from AM under drought conditions, resulting in an improved WUE_applied_, as well as higher thousand-kernel weight and harvest index compared to drought-stressed NM plants. Furthermore, due to the ability of AMF to deliver P and other nutritional compounds to their host plants, we expected to find lower C/N ratios due to higher N provisioning, as well as higher amounts of elementary P in AM plants. Finally, we expected to find a variation in the responses of wheat cultivars with respect to the used substrate.

## 2. Results

### 2.1. Water Status of NM Plants

Relative changes in hydration status were tested in a few NM plants with patch pressure clamps over time to gain a general insight into the effects of the irrigation treatment on water status. Generally, the curves of patch pressure (*P*_p_) showed different patterns for plants of the three water regimes (Figure 1). For all plants, the maximum *P*_p_ was reached around noon and the minimum was reached at night. CTR plants had a similar *P*_p_ pattern, with a mean patch pressure of 112.5 kPa, with increasing variation from 63 dps onwards. Plants subjected to CD treatment showed a consistently low *P*_p_ (mean = 79 kPa) over the entire measuring period. For CTR and CD plants, irrigation events had only minor to no impact on the course of the *P*_p_ curve. In contrast, the PD plants had the most variable pattern in *P*_p_ (mean = 107 kPa), which increased during the drought period and dropped after the first watering event. In the second drought interval, the increase of *P*_p_ until irrigation was less pronounced than in the first one.

### 2.2. Total Root Length Colonization

The total root length colonization was determined from subsamples of the roots collected at the final harvest. Mycorrhizal structures were present in all AM1 and AM3 samples, but not in NM plants (Figure 2). Mainly hyphae were found, but only a few vesicles and no arbuscles. Both irrigation and substrate had a significant impact on total root length colonization in both wheat cultivars. With increasing drought stress, TRLC decreased; CTR plants showed the highest root colonization (range 13–20%) and across AM groups, were on average 1.2 (MIS) and 1.8 (CHA) times more mycorrhized than CD plants, while PD plants showed the least mycorrhizal structures (Figure 2). Roots of AM3 plants were on average 1.7 (MIS) and 2.2 (CHA) times more mycorrhized than AM1 plants within the same irrigation group.

### 2.3. Applied Water Use Efficiency and Plant Biomass

The WUE_applied_ (dry aboveground biomass divided by cumulative irrigation amount each pot received until harvest) was significantly influenced by irrigation and substrate at both time points, in MIS plants at T1 as individual terms and otherwise in their interaction (Figure 3). At T1, the WUE_applied_ was highest in CTR NM and CD plants in both cultivars, while PD plants showed the overall lowest WUE_applied_. In CTR plants of both cultivars, the WUE_applied_ was higher in NM than in mycorrhized plants. In contrast, the CD and PD plants of the CHA cultivar showed, by tendency, a higher WUE_applied_ when being mycorrhized at T1. At T2, CTR and CD plants showed a generally lower WUE_applied_, whereas PD plants remained within the same range. For MIS plants, the pattern between the different treatments remained similar to T1. In the CHA cultivar, CD NM plants were within the same range as AM plants at T2. The total aboveground biomass was similarly influenced by the interaction of irrigation and substrate (Appendix A). Plants exposed to drought had a much lower biomass than CTR plants. The presence of AMF led to a reduction in biomass, especially in CTR plants, while the effect of substrate was less pronounced under water shortage.

### 2.4. Harvest Index

At T1, the harvest index (dry ear biomass/dry total aboveground biomass) was only significantly influenced by the irrigation treatments (Figure 4). Averaged over all plants within each irrigation group, CD plants had the highest harvest index in both wheat cultivars regardless of substrate, being 1.4 (MIS) and 1.2 times (CHA) higher in CD compared to CTR and PD plants. At T2, the pattern for the harvest index was noticeably different from T1; both irrigation and substrate had significant effects, as individual terms in MIS and in their interaction in CHA plants. The harvest index of the CTR and CD plants was comparable and on average 1.3 (MIS) and 1.2 (CHA) times higher than in PD plants. Furthermore, within the CHA cultivar, NM plants of the CD and PD irrigation regime showed a lower harvest index compared to mycorrhized plants within the same irrigation group.

### 2.5. Thousand-Kernel Weight

The thousand-kernel weight (total weight of grains per pot/number of grains per pot) in MIS plants was significantly influenced by the substrate but not by the irrigation treatment (Figure 5). CTR NM and PD NM plants had the highest values and were also higher compared to AM plants within the same irrigation groups, while the CD plants had comparable thousand-kernel weights across substrates. For CHA plants, both irrigation and substrate individually had a significant effect on the thousand-kernel weight. Within substrate groups, thousand-kernel weights were lower in CD compared to CTR and PD plants and lower in AM than NM plants.

### 2.6. Contents of C, N and C/N Rratio

Overall, the C content was just marginally influenced by the irrigation (Appendix A), being highest in both vegetative biomass and ears of CTR plants in both cultivars. In contrast to C, N content was significantly influenced by both irrigation and substrate, in MIS plants as individual factors and in CHA plants in their interaction (Appendix A). Generally, ears had a slightly higher N content than the vegetative biomass. Over both cultivars and plant parts, N contents increased along with the stress severeness, whereas CTR plants always had the lowest N contents, and PD plants, on average, had the highest. In both cultivars, N contents were, on average, 1.2 times higher in NM plants subjected to the CD and PD treatments compared to AM plants.

Overall, the C/N ratio showed a similar pattern across plants exposed to different irrigation regimes regardless of cultivar or plant part; CTR plants had the highest C/N ratios, followed by CD and then PD plants (Figure 6). In MIS plants, the C/N ratios of both the vegetative biomass and the ears were significantly influenced by irrigation and substrate, with NM plants having lower C/N ratios than mycorrhized plants. A similar pattern was found for the ears of CHA plants, while the C/N ratio of the vegetative biomass of CHA plants was significantly influenced by the interaction of irrigation and substrate in this cultivar. Here, NM CTR plants had higher C/N ratios than AM plants, whereas CD and PD plants showed the opposite pattern. The C/N ratio of the ears was altogether lower than that of the vegetative biomass.

### 2.7. P Content

The content of elementary P in the vegetative biomass and the ears in both cultivars was significantly influenced by the irrigation treatments, only (Figure 7). PD plants had, on average, always the highest P content compared to CD and CTR plants.

### 2.8. Mycorrhizal Growth Dependency

The calculated mycorrhizal growth dependency (100*(aboveground dry biomass of AM plant – averaged aboveground dry biomass of NM plants)/aboveground dry biomass of NM plants) revealed an overall rather negative response to AM in plants of all water regimes (Appendix A). Plant responsiveness to AM was significantly shaped by the irrigation, but not by different AM substrates. The negative effect of AM was particularly profound in CTR plants of both cultivars. In the MIS cultivar, CD plants were as severely influenced as CTR plants. Within the CHA cultivar, CD plants showed, on average, a neutral mycorrhizal growth dependency, while the AM effect was negative for plants of the PD treatment. Across all groups, only PD plants of the MIS cultivar growing with AM1 showed a positive responsiveness towards AM.

## 3. Discussion

In the present study, the irrigation treatment clearly affected the growth and resource allocation of spring wheat plants. More water at once at a lower frequency (i.e., PD) led partly to more severe effects on WUE_applied_ and HI than less water at a higher frequency (CD), while well-watered CTR plants showed a high HI at the latest harvest and had the highest C/N ratios over all water regimes. Against our expectation, the presence of AMF was less beneficial; rather, it reduced nutritional levels of plants and manifested in a negative mycorrhizal growth dependency in most plants.

The measured absolute *P*_p_ in NM plants was clearly influenced by the irrigation regime. The higher values of *P*_p_ in CTR compared to drought-stressed plants indicate lower turgor within these plants. The increase in *P*_p_ in some of the CTR plants, especially in the second half of the measurements, may have been caused by leaf senescence and/or a thinning of the leaf induced by the clamps [48]. The constantly low *P*_p_ observed in CD flag leaves might indicate that the plants had adapted well to the low water availability after 30 days, and that they were able to maintain a stable turgor. The different patterns of *P*_p_ time courses in CD versus PD plants emphasize the impact of irrigation volume and frequency, as all plants received the same amount of water in eight days, but either in high (CD) or low (PD) frequency. With low frequency, *P*_p_ gradually increased between watering events, similar to what was previously observed in wheat subjected to water shortage [48]. The pattern of *P*_p_ recordings gives insight into how soil water content (SWC) and watering frequency influence the plant hydration status, being shown to correlate well with leaf water potential and transpiration [48].

Overall, the total root colonization with mycorrhiza was relatively low, especially compared to forbs, where colonization rates up to 90% can be found [25]. However, our results are in line with other studies on Poaceae, including wheat, in which root colonization is usually quite low, and lower than in forbs [32,36,49]. Several plants of the Poaceae family have been shown to have a low mycorrhizal growth dependency, which may be alleviated by a more complex fine root structure [50]. The total root colonization in the present experiment was particularly low in drought-stressed compared to CTR plants. Such reduced total root length colonization under drought conditions has been observed both in the laboratory and under field conditions in wheat [36,51]. Under drought, plants may not deliver as much surplus C to the mutualist as needed by the AMF for their development and for the successful establishment of AM. However, a mixture of three AMF species led to a much higher total root length colonization than when plants were inoculated with just one AMF species in the present study. In other studies on wheat, the root colonization varied strongly among wheat cultivars and the used fungus or fungal community [32,52]. Using a multiple species inoculum in well-watered spring wheat (cultivar: 1110) increased grain yield [53]. However, more studies are needed to test the effects of multi-species inocula or native AMF communities on wheat [54].

The WUE_applied_ was, on average, higher in wheat plants grown under CD conditions compared to PD conditions, both at T1 and T2. When soil moisture decreases, plants close their stomata to prevent further water loss via transpiration and thereby maintain their internal water status [55]. Under mild water deficit, this adjustment can lead to an improved WUE [10], indicating a more effective use of water for biomass production. A similar pattern for WUE_applied_ in plants kept under CD versus PD has been found previously in wheat of the cultivar Tybalt [56], and may thus be characteristic for different wheat cultivars in general. The WUE_applied_ in CTR plants at T1 was surprisingly high and, on average, similar to that of CD plants in the present experiment, although CTR plants had, in principle, unlimited access to water. In contrast, plants of the crop savoy cabbage showed a significantly lower WUE_applied_ under well-watered conditions than under drought stress [57], highlighting that those responses may be highly species-specific and depend on the given irrigation treatment. Mycorrhizal structures can contribute to the water capacity of soil over their mycelium and excretion of glomalin [58]. However, against our expectation, AMF presence did not significantly improve the WUE_applied_ of plants exposed to drought, regardless of the substrate (AM1 *versus* AM3). In contrast, particularly in well-watered CTR plants, the WUE_applied_ was much lower in AM compared to NM plants. This effect may be caused by an altered stomatal conductance induced by AM, leading to higher transpiration rates ([59,60] Quiroga, 2018 #110).

The higher WUE_applied_ in CD compared to PD plants in the present experiment was reflected in the harvest index, being also higher in CD plants. Interestingly, for CTR plants, the harvest index was much lower than that of PD plants at T1, while it was more similar between CTR and PD plants at T2. Such differences between plants of the different irrigation treatments over time may be related to differences in development. In order to cope with water deficit, one strategy of plants is to accelerate the reproductive phase before the stress gets too severe [61]. Here, CD plants may have been already further developed than CTR plants at T1 (although having overall a much lower aboveground biomass), while at T2, CTR plants may have reached a similar stage. These changes over time highlight that it is important to investigate the effects of irrigation treatments on plant performance traits at different time points across plant development.

Although PD plants showed a lower harvest index than CTR and CD plants at T2, the thousand-kernel weight at T2 was not affected by irrigation, at least in MIS plants, and in the CHA cultivar, PD plants also reached a relatively high kernel weight. During the vegetative development of crops, it is decisive when water shortages occur [62]. The number of ears and grains can, for instance, be impaired if drought occurs during anthesis [63]. In the present experiment, one or more watering events may have taken place just at the right time point, enabling plants of the PD treatment to use the water for grain development effectively. Moreover, fertilization of plants occurred on days at which all groups were watered, enabling all plants to absorb nutrients. As PD pots received a larger volume of water on these days, the substrate in these pots may have been wet for a longer time, allowing for more nutrient uptake.

In response to AMF, PD plants in particular, but also CD plants (at least of CHA), responded positively to AMF in terms of a higher harvest index. This potential growth benefit was also observed in AM wheat plants subjected to CD and PD stress in a previous experiment [19]. However, the presence of AMF seemed to have a negative impact on the thousand-kernel weight in plants of almost all irrigation groups (except for MIS plants of the CD treatment). Indeed, on average, PD AM plants produced 1.3 times more grains than NM plants, but the grain weight did not increase proportionally (data not shown). Other experiments revealed that AM led to a higher yield in wheat [53,64]. At the same time, water shortages have an impact on later grain filling, and hence, the overall yield [62].

At T1, when plant material was harvested for C and N analysis, the grains had reached watery ripeness and the biomass of ears showed overall lower C/N ratios, due to higher N contents, than vegetative parts, likely due to enhanced resource allocation into the reproductive organs. Both drought and substrate significantly influenced the C/N ratios of all plant parts in both cultivars. C/N ratios decreased with increasing drought stress in both vegetative biomass and ears, emphasizing that the irrigation frequency also affects this trait. Likewise, a lower C/N ratio was found in drought-stressed compared to well-watered plants of savoy cabbage [57]. The stomatal closure in response to low soil moisture [10] can lead to a limited CO_2_ assimilation, and thus, lower C allocation in foliar tissues. At the same time, investment in growth is reduced under drought, as also seen in the present experiment, allowing for a relatively higher N-uptake [65].

In contrast to our hypothesis and common findings in many species (e.g., meta-analysis [66]), the C/N ratio was higher in AM than in NM plants in both vegetative biomass and ears, particularly in CD and PD plants and in plants of the MIS cultivar also in CTR plants. C/N ratios can increase due to enhanced contents of C and/or lower N. Higher foliar C content could potentially be caused by AM-altered stomatal conductance, and hence, an improved CO_2_ assimilation [59,60]. Moreover, AM has been shown to increase chlorophyll content [67], which would also contribute to higher foliar C content. However, this is not what we observed, as in our experiment, the C content of plant material was only moderately affected, but N was rather drastically reduced in presence of both AM substrates. The reduced N content of AM plants may be explained by the need of N for the foundation of the relationship itself, as not only the plant, but also the fungal partner, need N-containing sources to grow [30]. Under N-limited conditions, both plant and AMF can become competitors for N and the relationship may shift into parasitism [42]. Although the fertilizer used in the present experiment did contain N, the absolute N amount may have been too low to support both the wheat plants and the AMF. Another explanation for lower N contents in AM plants may be a potential re-allocation of N to the roots, as has been shown for two other spring wheat cultivars (Vinjett and 1110) [53].

For the content of elemental P, we had expected to find the highest amounts in well-watered CTR plants, since they were likely not inhibited in their nutrient uptake. In addition, we had expected higher P contents in plants growing with AMF, as the fungi can improve the P uptake of the roots [33,54]. Surprisingly, only irrigation had a significant effect on P content of both the vegetative biomass and ears, with the P content being highest in PD plants. This is in contrast to other findings, for example, for bread wheat, in which higher levels of P were found in shoots of AM plants in three different cultivars [46], regardless of drought [64]. The high P levels in our experiment, particularly in PD plants, may be explained by short-term waterlogging effects directly after watering. At least for a grass community *of Paspalum dilatatum,* it has been shown that waterlogging of these plants led to an overall increase in P-content in above- and belowground biomass, even though the total plant biomass was not higher in waterlogged plants [68]. High water amounts can lead to mobilization and solubilization of P-containing compounds and increase their availability for plants [69,70], which is particularly advantageous in P-limited soils [68].

The results from this greenhouse experiment provide insights on how common wheat cultivars could perform under future climate change scenarios. Drought led to a suppressed growth, and particularly longer drought cycles caused even more severe shifts in generative plant development, finally leading to a reduction in the harvest index, an agriculturally important trait. Although in other studies, alleviating effects of AM on drought stress in wheat were found [64,67,71], the presence of AMF does not necessarily lead to improved plant performance under water scarcity. The low responsiveness to AMF, at least in some traits, and a negative mycorrhizal growth dependency, shown in reduced total biomass, may be related to the fact that wheat is intensively selected for other yield-related traits and may have thereby lost key signals to enter a close mutualistic interaction with AMF [72], at least in some cultivars [47].

## 4. Materials and Methods

### 4.1. Plant Cultivation and Experimental Set-Up

We used a full-factorial experimental design with two wheat cultivars [Mistral (MIS) and Chamsin (CHA)], growing them in three different substrates [containing either one (AM1, *Rhizoglomus irregulare*) or a mixture of three AMF species (AM3, *R. irregulare, Funneliformis mosseae, F. caledonium*) versus a non-mycorrhized control (NM)] under three irrigation regimes [well-watered control (CTR) versus continuous drought (CD) versus pulsed drought (PD)]. Continuous drought conditions were realized by providing less water at high frequency, whereas pulsed drought resembled higher water amounts at low frequency (Figure 8; for details see below). The experiment was carried out in an air-conditioned greenhouse chamber with long day light conditions (16:8 light:dark, on average 21 °C and photosynthetically active radiation of 183 µmol m^−2^ s^−1^). Wheat plants of the cultivars MIS and CHA (KWS SAAT SE & Co. KG, Einbeck, Germany) were grown in 2 L pots (11.3 ×11.3 × 23 cm; n = 108 pots/cultivar) in a mixture of sand and soil (steamed at 90 °C; soil: Fruhstorfer Pikiererde, HawitaGroup, Vechta, Germany). The wheat cultivars were chosen based on their cultivation and yield properties. Both cultivars reach a medium height (~50 cm) compared to other common cultivars, allowing for easy handling during the experiment (weekly rotation) and in the greenhouse in general. MIS and CHA are similar in most growth properties but do slightly differ in their yield quality, with MIS having more grains per ear and a higher thousand-kernel weight compared to CHA. To imitate field conditions, two plants were grown per pot in competition at a 10 cm distance. The substrate of the non-mycorrhizal (NM) control group was mixed with 200 mL autoclaved (at 120 °C) sand (INOQ GmbH; NM; n = 72 pots). For AM plants, the potting substrate was either mixed with 200 mL of a sand-based inoculum containing a single fungus (*Rhizoglomus irregulare*; INOQ GmbH, Schnega, Germany; AM1, n = 72 pots) or three fungi (*R. irregulare*, *Funneliformis mosseae*, *F. caledonium*; INOQ GmbH; AM3; n = 72 pots). The final substrate had a 2:1 sand:soil ratio in all treatments. With subsamples of these substrates, the initial SWC was determined. Each pot including the substrate was then individually watered near field capacity (SWC of 18%; determined in preliminary experiments), based on the pot mass (balance: Sartorius, Göttingen, Germany). To ensure proper initial growth and facilitate initiation of mycorrhization in AM plants, all pots were kept at these well-watered conditions for the first 22 days. Therefore, plants were watered every other day up to an SWC of 18%, based on the average weight of a random subsample of 16 control pots. Each pot had small holes at the ground allowing for draining and was placed on an individual dish holding all water.

At 23 dps, pots were randomly assigned to different irrigation regimes (n = 12 pots per wheat cultivar, irrigation and substrate regime, respectively). The well-watered control plants (CTR) were further watered to an SWC of 18% every other day. For continuous drought (CD) and pulsed drought (PD), pots were kept unwatered from 23 dps until they reached an SWC of 8%, which was the case at 26 dps. Starting at 26 dps, CD plants received only 40% of the water amount that the CTR plants received every other day. The PD group was watered only every eight days and received the cumulated amount of water that CD plants received within this time (Figure 8). For sufficient nutrient supply, all pots were fertilized at 19 and 34 dps with 2 g of a solid long-term, P-free mineral fertilizer (Floranid N-P-K 14-0-19, containing 3% Mg, 11% S, and traces of B, Cu, Fe, Mn, and Zn; Compo Expert, Münster, Germany).

### 4.2. Water Status Measurement with ZIM Probes

To gain a general insight into the hydration status of the wheat plants exposed to the three irrigation treatments, we measured relative changes in leaf turgor pressure via non-invasive pressure clamps (ZIM probes, Yara International ASA, Oslo, Norway). One clamp was attached to the flag leaf of each of three plants (two CHA, one MIS) per irrigation regime at 52 dps and remained on the same position until 67 dps to record the patch pressure (*P*_p_). Only NM plants were used for this purpose, due to a limitation in available pressure clamps.

### 4.3. Plant Harvest and AMF Quantification in Roots

To examine how the irrigation regimes and substrates influence wheat development, one subset of plants (n = 6 pots per wheat cultivar, irrigation and substrate treatment) was harvested at 67 dps (T1, grains water ripe) and the other at 113 dps (T2, grains). At each timepoint, aboveground plant material was cut and separated by vegetative biomass (stems and leaves) and reproductive parts (ears). The plant material was dried (40 °C, 96 h, oven UT12, Heraeus, Germany) and weighed. The WUE_applied_ was calculated by dividing the dry aboveground biomass per pot by the cumulative irrigation each pot received until T1 or T2. The harvest index was calculated by dividing the dry ear biomass by the dry total aboveground biomass of the two plants in each pot at T1 and T2. The thousand-kernel weight was determined at T2 by dividing the total kernel yield per pot (in g) by the number of kernels and multiplying by 1000. To evaluate the influence of mycorrhizal substrate AM1 and AM3 on plant growth, the mycorrhizal growth dependency was calculated for each irrigation group and substrate [100*(aboveground dry biomass of AM plant – averaged aboveground dry biomass of NM plants)/aboveground dry biomass of NM plants)] at T2. To determine the total root length colonization by AMF at T1, two subsamples of belowground biomass were taken per pot with a soil sampler (2.8 cm i.d.), sampling vertically through the substrate. Root samples were cleaned from soil and representative subsamples were bleached in 10% KOH (15 min at 95 °C), dyed with an ink solution (royal blue, Ink 4001, Pelikan Group GmbH; 1:1:8 ink:acetic acid:water; 20 min at 90 °C) and conserved in a 4:2:1 mixture of 90% lactic acid:89% glycerin:water. The total root length colonization was determined with the grid-line intersect method [73], counting any AMF structure within the roots in 200 intersects per sample. Mycorrhization was not evaluated at T2 because preliminary experiments showed that, at this time point, many mycorrhizal structures had already degraded.

### 4.4. Determination of C, N and P Content

Dried plant material of vegetative aboveground biomass (leaves and stems) as well as reproductive biomass (ears) from T1 were homogenized, and subsamples used for the following chemical analyses. Contents of C and N within the plant tissues were analyzed via a C/N analyzer (Vario MICRO Cube; Elementar Analysensysteme, Hanau, Germany) after high-temperature combustion. Orthophosphate was determined by high-temperature oxidation, followed by photometrical quantification as according to Watanabe and Olsen [74]. Plant material was mineralized by incineration at 500 °C and subsequent digestion of 1 mg ash with 1 mL 10% nitric acid. The extracts were diluted (1:8) with bi-distilled water and photometrically analyzed at 880 nm via flow injection analysis (FIA-Laboratory II, MLE GmbH, Dresden, Germany).

### 4.5. Statistical Analyses

All statistical analyses were carried out using R [75]. If not stated otherwise, linear models (LMs) were calculated for all responses and separately for each cultivar and time point. We used visual inspection following Zuur, et al. [76] to check model variance homogeneity and the normal distribution of residuals for all calculations. Separate models were calculated for the two cultivars, MIS and CHA, to avoid overfitting of models. To test the response of total root length colonization (arcsine- and square-root-transformed), a LM with the factorial predictors irrigation (CTR, CD and PD) and mycorrhiza treatment (AM1 and AM3) was performed. The responses of total aboveground biomass, harvest index, thousand-kernel weight, WUE_applied_, C and N content (absolute, in percentage), C/N ratio and P-content initially comprised the factorial predictors irrigation (CTR, CD and PD) and substrate (NM, AM1 and AM3). For mycorrhizal growth dependency, the first LM included the factorial predictors irrigation (CTR, CD and PD) and mycorrhiza treatment (just the two levels, AM1 and AM3). Single term deletion was used for non-significant terms in all models and only significant terms were kept, using the package *mass* [77].

## Figures and Tables

**Figure 1 plants-11-02187-f001:**
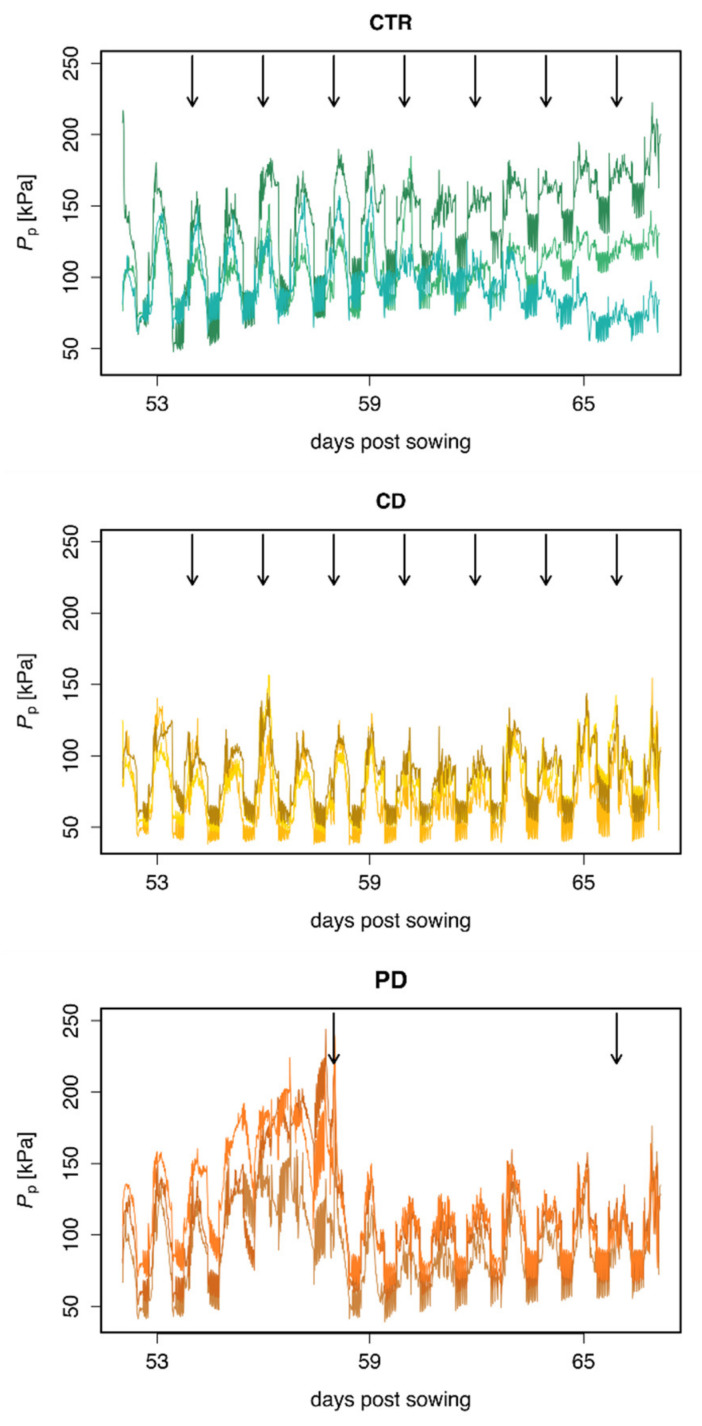
Continuous patch pressure (*P*_p_) recordings of flag leaves from non-mycorrhized wheat plants (*Triticum aestivum*) shown over time [days post sowing (dps)] recorded with ZIM-probes. Cultivation occurred under different irrigation conditions [control (CTR), continuous drought (CD) and pulsed drought (PD)]. Measurements started 52 dps (two days after last watering event for PD plants) and were recorded until 67 dps. Arrows indicate watering events (see Figure 8 for experimental set-up). Higher *P*_p_ values indicate a lower turgor. During the measuring period, the pressure clamps were not re-clamped.

**Figure 2 plants-11-02187-f002:**
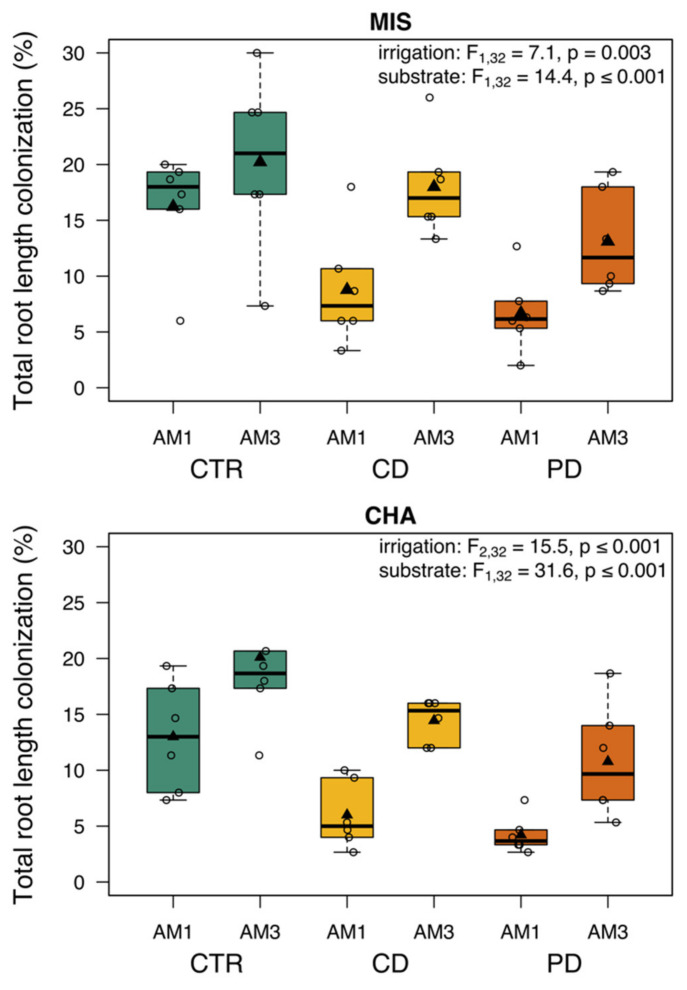
Total root length colonization of roots from two wheat cultivars [*Triticum aestivum*; Chamsin (CHA), Mistral (MIS)] 67 days post sowing. Cultivation occurred under different irrigation conditions [control (CTR), continuous drought (CD) and pulsed drought (PD), factor: irrigation]. Plants were either mycorrhized with a single fungus (AM1; *Rhizoglomus irregulare*) or with a mixture of three fungi (AM3; *R. irregulare*, *Funneliformis mosseae*, *F. caledonium*) (factor: substrate). Data are given as box–whisker plots with interquartile ranges (IQR; boxes) including medians (horizontal lines), means (triangles) and whiskers (extending to the most extreme data points with maximum 1.5 times the IQR); raw data are given as open circles. The results of linear models are shown comprising the remaining significant terms after model simplification; n = 6.

**Figure 3 plants-11-02187-f003:**
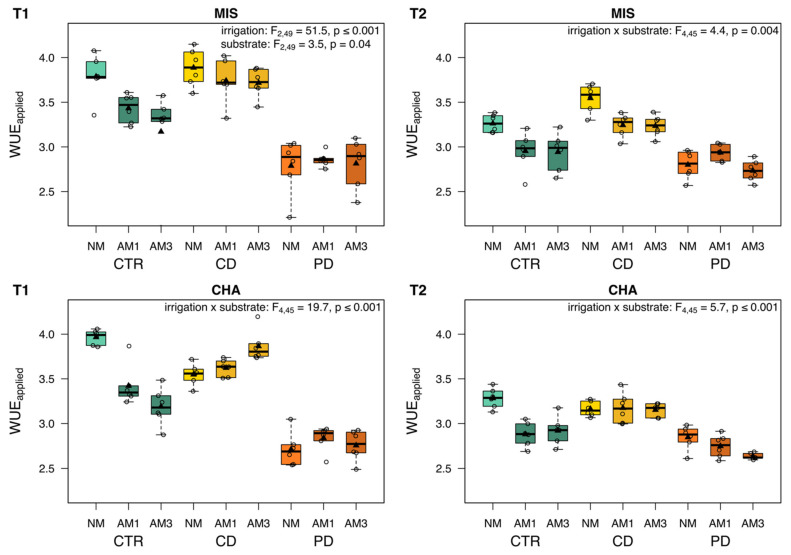
Applied water use efficiency (WUE_applied_) of two wheat cultivars [*Triticum aestivum*; Chamsin (CHA), Mistral (MIS)] 67 days (T1) and 113 days (T2) post sowing. Cultivation occurred under different irrigation conditions [control (CTR), continuous drought (CD), and pulsed drought (PD), factor: irrigation]. Plants were either non-mycorrhized (NM), mycorrhized with a single fungus (AM1; *Rhizoglomus irregulare*) or with a mixture of three fungi (AM3; *R. irregulare*, *Funneliformis mosseae*, *F. caledonium*) (factor: substrate). Data are given as box–whisker plots with interquartile ranges (IQR; boxes) including medians (horizontal lines), means (triangles) and whiskers (extending to the most extreme data points with maximum 1.5 times the IQR); individual values are given as open circles. The results of linear models are shown comprising the remaining significant terms after model simplification; n = 6.

**Figure 4 plants-11-02187-f004:**
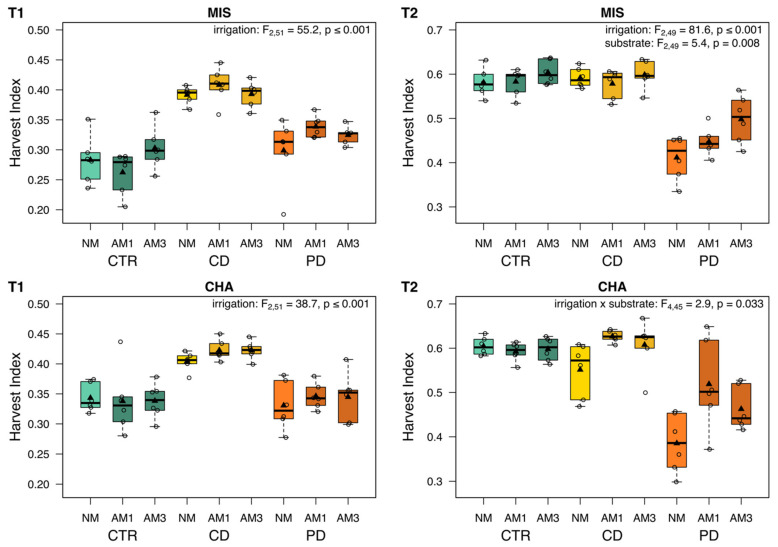
Harvest index of two wheat cultivars [*Triticum aestivum*; Chamsin (CHA), Mistral (MIS)] 67 (T1) and 113 (T2) days post sowing. Cultivation occurred under different irrigation conditions [control (CTR), continuous drought (CD) and pulsed drought (PD), factor: irrigation]. Plants were either non-mycorrhized (NM), mycorrhized with a single fungus (AM1; *Rhizoglomus irregulare*) or with a mixture of three fungi (AM3; *R. irregulare, Funneliformis mosseae*, *F. caledonium*) (factor: substrate). Data are given as box–whisker plots with interquartile ranges (IQR; boxes) including medians (horizontal lines), means (triangles) and whiskers (extending to the most extreme data points with maximum 1.5 times the IQR); individual values are given as open circles. The results of linear models are shown comprising the remaining significant terms after model simplification; n = 6.

**Figure 5 plants-11-02187-f005:**
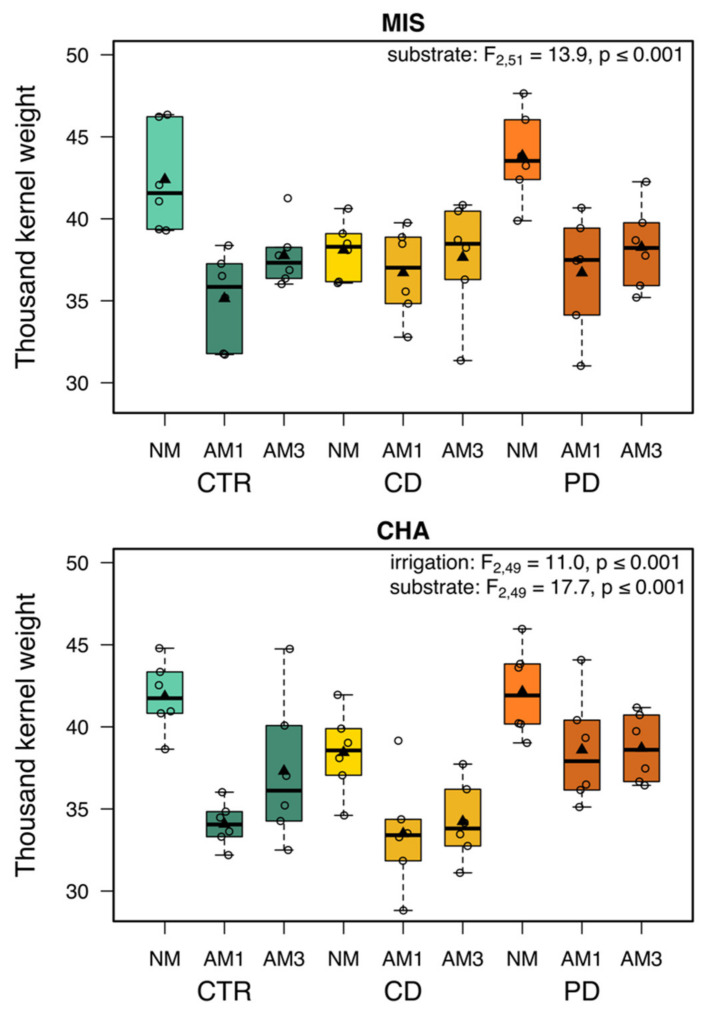
Thousand-kernel weight of two wheat cultivars [*Triticum aestivum*; Chamsin (CHA), Mistral (MIS)] 113 (T2) days post sowing. Cultivation occurred under different irrigation conditions [control (CTR), continuous drought (CD) and pulsed drought (PD), factor: irrigation]. Plants were either non-mycorrhized (NM), mycorrhized with a single fungus (AM1; *Rhizoglomus irregulare*) or with a mixture of three fungi (AM3; *R. irregulare*, *Funneliformis mosseae*, *F. caledonium*) (factor: substrate). Data are given as box–whisker plots with interquartile ranges (IQR; boxes) including medians (horizontal lines), means (triangles) and whiskers (extending to the most extreme data points with maximum 1.5 times the IQR); individual values are given as open circles.

**Figure 6 plants-11-02187-f006:**
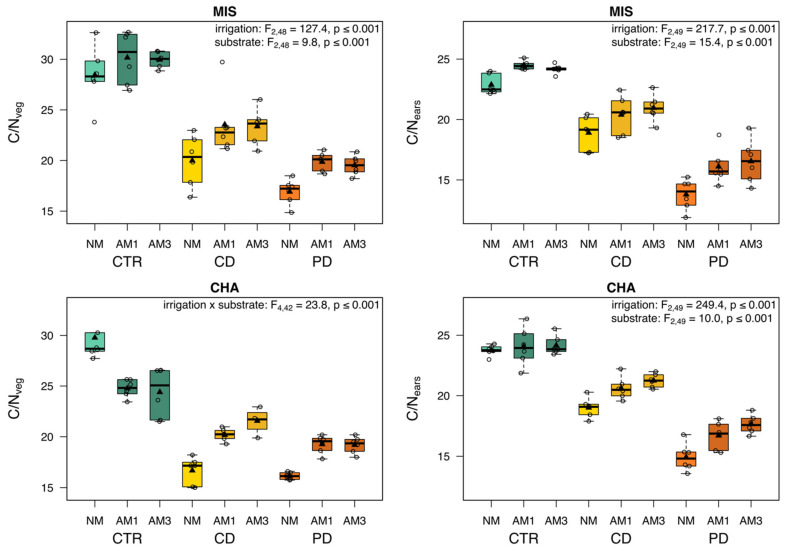
C/N ratio of vegetative biomass (stems and leaves) and ears from two wheat cultivars [*Triticum aestivum*; Chamsin (CHA), Mistral (MIS)] 67 days post sowing. Cultivation occurred under different irrigation conditions [control (CTR), continuous drought (CD) and pulsed drought (PD), factor: irrigation]. Plants were either non-mycorrhized (NM), mycorrhized with a single fungus (AM1; *Rhizoglomus irregulare*) or with a mixture of three fungi (AM3; *R. irregulare, Funneliformis mosseae, F. caledonium*) (factor: substrate). Data are given as box–whisker plots with interquartile ranges (IQR; boxes) including medians (horizontal lines), means (triangles) and whiskers (extending to the most extreme data points with maximum 1.5 times the IQR); individual values are given as open circles. The results of linear models are shown comprising the remaining significant terms after model simplification; n = 4–6.

**Figure 7 plants-11-02187-f007:**
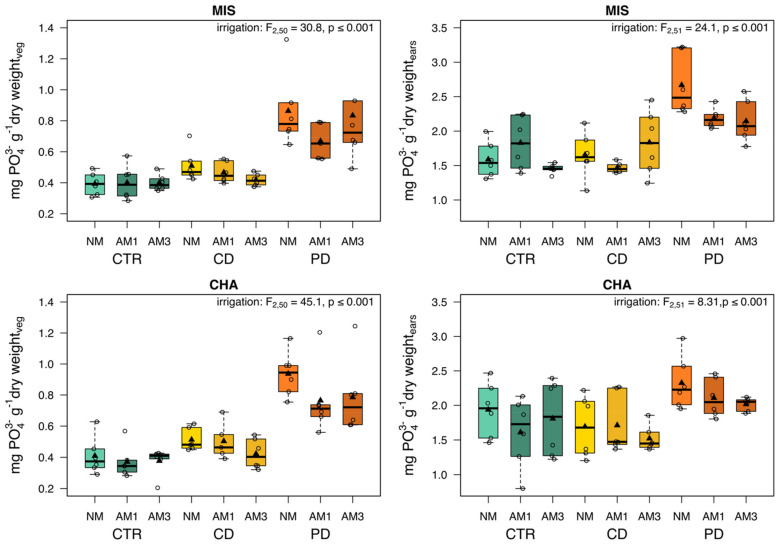
Content of elementary P of vegetative biomass (stems and leaves) and ears from two wheat cultivars [*Triticum aestivum*; Chamsin (CHA), Mistral (MIS)] 67 days post sowing. Cultivation occurred under different irrigation conditions [control (CTR), continuous drought (CD) and pulsed drought (PD), factor: irrigation]. Plants were either non-mycorrhized (NM), mycorrhized with a single fungus (AM1; *Rhizoglomus irregulare*) or with a mixture of three fungi (AM3; *R. irregulare, Funneliformis mosseae, F. caledonium*) (factor: substrate). Data are given as box–whisker plots with interquartile ranges (IQR; boxes) including medians (horizontal lines), means (triangles) and whiskers (extending to the most extreme data points with maximum 1.5 times the IQR); individual values are given as open circles. The results of linear models are shown comprising the remaining significant terms after model simplification; n = 5–6.

**Figure 8 plants-11-02187-f008:**
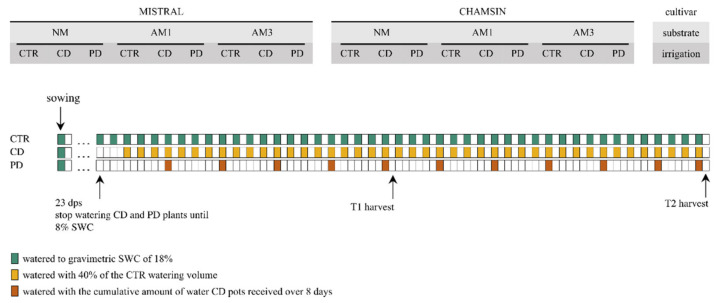
Experimental set-up. Plants of two wheat cultivars (*Triticum aestivum*; Chamsin, Mistral) were either non-mycorrhized (NM), mycorrhized with a single fungus (AM1; *Rhizoglomus irregulare*) or with a mixture of three fungi (AM3; *R. irregulare*, *Funneliformis mosseae*, *F. caledonium*) (factor: substrate). Plants were sown directly in pots (0 days post sowing, dps) and well watered every other day (days represented by boxes, watering events represented by colored fills) until 23 dps, when they were assigned to different irrigation regimes [well-watered (CTR); continuous drought (CD); pulsed drought (PD)] (factor: irrigation); SWC = soil water content.

## Data Availability

Data and code are provided in the Appendix A (raw data: Data_Manuscrip_Pons_Müller_2022.xlsx; code: Data_Analyses_PONS_MÜLLER.R).

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
