# Peer review of "Impacts of Drought Stress and Mycorrhizal Inoculation on the Performance of Two Spring Wheat Cultivars"

_plants, 2022, doi:10.3390/plants11172187_

Round 1

Reviewer 1 Report

Review Remarks on the manuscript: Impacts of Drought Stress and Mycorrhizal Inoculation on the Performance of Two Spring Wheat Cultivars. The concept of the manuscript was quite interesting, and the overall presentation of the manuscript is excellent. Besides, there are some minor points and scientific queries below mentioned need to be addressed in the manuscript:

·    Please check lines 54-55 and rephrase them.

·    In this statement: In contrast, some studies revealed positive effects of AM on the biomass production or harvest index; please mention plant names.

·    In line 76: species (composition) are shaped by environmental conditions such as soil moisture. Please include more environmental conditions or finish the sentence after environmental conditions.

·     If possible, please try to provide the values of F and p, on one side like right or left and top or bottom. At first glance, in some figures, it appeared that these values are not provided but after careful observation, I found them on another side.

·    I recommend adding recent literature, from the last two years to provide current insights, especially in the discussion part. Also, in the end, if authors can briefly include the acting mechanisms, it can enhance the scientific quality and readership of the article. 

Author Response

Review Remarks on the manuscript: Impacts of Drought Stress and Mycorrhizal Inoculation on the Performance of Two Spring Wheat Cultivars. The concept of the manuscript was quite interesting, and the overall presentation of the manuscript is excellent. Besides, there are some minor points and scientific queries below mentioned need to be addressed in the manuscript:

  •   Please check lines 54-55 and rephrase them.

REPLY: Thank you for your positive feedback. We rephrased the sentence (lines 53-55).

  •   In this statement: In contrast, some studies revealed positive effects of AM on the biomass production or harvest index; please mention plant names.

REPLY: This section focusses on wheat and mycorrhiza. We specified the sentence accordingly (line 66).

  •   In line 76: species (composition) are shaped by environmental conditions such as soil moisture. Please include more environmental conditions or finish the sentence after environmental conditions.

REPLY: We shortened the sentence by finishing the sentence after “environmental conditions” (line 78).

  •    If possible, please try to provide the values of F and p, on one side like right or left and top or bottom. At first glance, in some figures, it appeared that these values are not provided but after careful observation, I found them on another side.

REPLY: Thank you for this helpful suggestion. We altered our figures placing the F and p results always in the same corner (upper right) and by enlarging the figures using the entire page width.

  •   I recommend adding recent literature, from the last two years to provide current insights, especially in the discussion part. Also, in the end, if authors can briefly include the acting mechanisms, it can enhance the scientific quality and readership of the article. 

REPLY: We added two more recent studies on wheat and mycorrhiza to our manuscript [37, 46]. We discuss the potentially acting mechanisms at different places in the manuscript, see e.g. lines 294-296, 304-307, 320-321, 325-330. Since we did not perform physiological or genetic analyses in this study, we do not want to go beyond these suggestions to avoid being too speculative.

Reviewer 2 Report

The research topic is very interesting and may have great importance in wheat production. The manuscript contains many valuable results. These are my main comments on the manuscript entitled “Impacts of Drought Stress and Mycorrhizal Inoculation on the Performance of Two Spring Wheat Cultivars”. The authors examined the impacts of three irrigation and three treatments at different growth stages of two wheat cultivars’ main characteristics.

Overall the manuscript is clear, organized, and well-structured. The introduction was well written and gave a good background of the main idea of the study.

Results are statistically analyzed but the presentation should have improved. The figures are too small to see differences among treatments.

Specific comments

Figure 1: Please replace Materials and Methods.

Materials and Methods

Line 422: Please add the name of two cultivars here.

Line 423: Please explain the treatments and irrigations here.

Line 497: Please indicate the used temperature and length of drying, in addition to the type, producer, etc. of oven or drier.

I can suggest publishing this paper after a minor revision.

Author Response

The research topic is very interesting and may have great importance in wheat production. The manuscript contains many valuable results. These are my main comments on the manuscript entitled “Impacts of Drought Stress and Mycorrhizal Inoculation on the Performance of Two Spring Wheat Cultivars”. The authors examined the impacts of three irrigation and three treatments at different growth stages of two wheat cultivars’ main characteristics.

Overall the manuscript is clear, organized, and well-structured. The introduction was well written and gave a good background of the main idea of the study.

REPLY: Thank you very much for your positive feedback!

Results are statistically analyzed but the presentation should have improved. The figures are too small to see differences among treatments.

REPLY: We enlarged several figures by using the entire page size.

Specific comments

Figure 1: Please replace Materials and Methods.

REPLY: We moved the figure with the experimental set-up to the methods section and adjusted the figure numbers accordingly (original Figure 1 is now Figure 8).

Materials and Methods

Line 422: Please add the name of two cultivars here.

REPLY: We added the names and abbreviations of the cultivars (lines 409-410).

Line 423: Please explain the treatments and irrigations here.

REPLY: We added the information on the treatments and irrigations again (lines 410-414).

Line 497: Please indicate the used temperature and length of drying, in addition to the type, producer, etc. of oven or drier.

REPLY: We described the drying process in the methods section under the subsection “Plant Harvest and AMF Quantification in Roots” (line 476). We have now added the manufacturer of the oven (line 476).

I can suggest publishing this paper after a minor revision.

REPLY: We thank you for your feedback and hope that our improvements are sufficient.